# Reduced Levels of Brain-Derived Neurotrophic Factor Affect Body Weight, Brain Weight and Behavior

**DOI:** 10.3390/biology13030159

**Published:** 2024-02-29

**Authors:** Matthias Wilhelm Voigt, Jens Schepers, Jacqueline Haas, Oliver von Bohlen und Halbach

**Affiliations:** Institut für Anatomie und Zellbiologie, Universitätsmedizin Greifswald, Friedrich-Loeffler Str. 23c, D-17489 Greifswald, Germany

**Keywords:** BDNF, central nervous system, knockout, obesity, behavior

## Abstract

**Simple Summary:**

Neurotrophins are growth factors that help the brain grow and function well. One of them is called brain-derived neurotrophic factor (BDNF). BDNF affects how much we weigh and how well we learn and remember things. There are mice with reduced levels of BDNF. Mice have been developed that do not express BDNF at all, but they die soon after they are born. So, mice that have only half of the normal BDNF level and mice that have very low BDNF levels in some brain cells, but normal BDNF levels in other brain cells, have been generated. Furthermore, it is possible to generate new mouse lines by breeding these two types of mice together. These new mice have very little BDNF in their brain. They are alive, but they weigh more and have smaller brains than normal mice. They also act differently, especially in how they move.

**Abstract:**

Neurotrophins, which belong to the family of growth factors, not only play crucial roles during development but are also involved in many processes in the postnatal brain. One representative of neurotrophins is brain-derived neurotrophic factor (BDNF). BDNF plays a role in the regulation of body weight and neuronal plasticity and is, therefore, also involved in processes associated with learning and memory formation. Many of the studies on BDNF have been carried out using BDNF-deficient mice. Unfortunately, homozygous deletion of BDNF is lethal in the early postnatal stage, so heterozygous BDNF-deficient mice are often studied. Another possibility is the use of conditional BDNF-deficient mice in which the expression of BDNF is strongly downregulated in some brain cells, for example, in the neurons of the central nervous system, but the expression of BDNF in other cells in the brain is unchanged. To further reduce BDNF expression, we crossed heterozygous BDNF-deficient mice with mice carrying a deletion of BDNF in neurofilament L-positive neurons. These offspring are viable, and the animals with a strong reduction in BDNF in the brain show a strongly increased body weight, which is accompanied by a reduction in brain weight. In addition, these animals show behavioral abnormalities, particularly with regard to locomotion.

## 1. Introduction

Brain-derived neurotrophic factor (BDNF) is a member of the neurotrophin family. Neurotrophins not only play important roles during development [1,2] but are also involved in postnatal processes, e.g., in the maintenance and function of the nervous system [3]. BDNF can signal through the high-affinity receptor trkB as well as through the pan-neurotrophin receptor p75NTR [4]. In humans, the BDNF Val66Met polymorphism can contribute to increased body weight [5], major depression [6], reductions in the size of brain areas [7] and declines in memory capacity [8]. Based on these observations, especially concerning the involvement of reduced levels of available BDNF in major depression and memory capacity, it is not surprising that BDNF is thought to play a fundamental role in neuronal plasticity. Neuronal plasticity refers to the ability of neurons to change their structure and function in response to changes in the environment or in response to stimuli. This process is critical for learning and memory, and is also important for the development and maintenance of normal brain function. Structural changes related to neuronal plasticity can be observed at the level of dendritic spines or, especially in the hippocampus, as changes in adult hippocampal neurogenesis.

Unfortunately, a loss of functional nerve growth factor (NGF), neurotrophin-3 (NT-3) or BDNF genes results in severe neuronal deficits and early postnatal death [9]. For example, mice lacking BDNF mainly die during the second postnatal week [10]. Nevertheless, mice that are heterozygous for BDNF survive into adulthood and display BDNF levels that are reduced by about half [11]. Another approach for investigating the effects of a deletion of BDNF in the central nervous system is the generation of conditional BDNF-knockout mice that lack BDNF in neurofilament light chain (NF-L)-expressing cells [12]. Such conditional BDNF-deficient mice lack BDNF mainly in pyramidal neurons [12], but BDNF levels in other cell types remain unaltered. The crossing of mice with a heterozygous deletion of BDNF with mice that display conditional deletion of BDNF in specific cell types of the brain allows the generation of mice that display a further significant reduction in the brain but might survive into adulthood. By following this approach, we recently generated mice that displayed a strong reduction in brain BDNF but survived into adulthood. In a first comparative study, we monitored these mice for about one year in terms of food intake and body weight development. The gross morphology of age-matched adult mice was analyzed as well as the hippocampal BDNF levels. Moreover, age-matched (3 months, 6 months and 9 months of age) heterozygous BDNF-knockout mice, NF-L conditional BDNF-knockout mice and heterozygous BDNF/NF-L conditional BDNF-knockout mice were subjected to different behavioral tests.

## 2. Materials and Methods

Breeding pairs of conditional BDNF-knockout (KO) mice were obtained from the laboratory of Michael Sendtner (Würzburg, Germany) and bred in the animal facilities of the University Medicine Greifswald. These conditional BDNF KO mice were generated by crossing mice carrying a bdnf exon V, which is flanked by two loxP sites on one allele and a neomycin cassette in the coding region of exon V on the second allele, with mice expressing cyclic recombinase (CRE) under the control of the promoter of neurofilament light chain (NF-L). For details regarding the generation of these mice, see [12]. Breeding pairs of BDNF +/− mice were also obtained from the laboratory of Michael Sendtner (Würzburg, Germany), and BDNF +/− mice were bred in the animal facilities of the University Medicine Greifswald. The generation of these mice was initially described by Korte and coworkers in 1995 [11]. By crossbreeding these two different BDNF-deficient mice, we generated mice that are heterozygous for BDNF in most organs, but with a further reduction in brain BDNF. These mice were termed −/fl,Cre+. In detail, the following mouse lines were analyzed: mice without a deletion in the BDNF system (+/+,Cre+ as well as fl/fl/Cre−), heterozygous BDNF mice (+/+,Cre+), mice lacking BDNF in NF-L-positive neurons (fl/fl,Cre+), and mice with a strong reduction in brain BDNF and heterozygous for BDNF in the periphery (−/fl,Cre+). The animals were kept in a 12 h day/night cycle with food and water access ad libitum.

For body weight analysis, the animals were weighed once a week using a scale (KS 36 Beurer, Ulm, Germany). To determine the weekly feed quantity, a beaker was placed on a standard kitchen scale and zeroed. Food was then poured into the beaker, and a weekly average was calculated from the difference between the amount of food added in the previous week and the amount in the trough on the day of measurement. For determining the brain wet weight of the mice, the animals were euthanized and then perfused via the heart, first with chilled phosphate-buffered saline (PBS) and then with 4% paraformaldehyde (PFA) solution. Their brains were explanted and post-fixed in 4% PFA. Brain weight was determined using a scale (KS 36 Beurer, Germany). Measurements were repeated three times for each brain. Serial coronal brain sections (section thickness of 30 μm) were created using a vibratome, and the sections were transferred into the wells of a 24-well plate (Falcon^®^ 24-well plate, Corning Inc., Corning, NY, USA) filled with 20% ethanol. Thereafter, the sections were mounted on Superfrost slides (R. Langenbrinck GmbH, Emmendingen, Germany) and dried. Next, the sections were rinsed in aqua destillata (A. dest.) for rehydration. For antigen retrieval, the sections were incubated with 10 mM sodium citrate buffer for 10 min in a microwave at 800 W. Subsequently, the sections were incubated in a solution containing normal goat serum (NGS) and Triton X-100 for 2 h at room temperature in a humidity chamber. Then, the sections were washed once with PBS and transferred to a solution containing the primary antibody (polyclonal anti-BDNF, Preprotech/ThermoFisher Scientific, Dreieich, Germany (500-P84)) for 12 h at 4 °C. After three washes in PBS, the sections were incubated with a solution containing Alexa Fluor^®^ 488-AffiniPuren Goat Anti-Rabbit IgG antibodies (DIANOVA, Hamburg, Germany) diluted 1:400 in the presence of 5% NGS and 0.1% Triton X-100 for 2 h at 4 °C. The slides were then washed three times in PBS. Counterstaining with 4’,6-diamidine-2-phenylindole (DAPI; diluted 1:10,000 in A. dest.) was performed to visualize the cell nuclei. After a further wash step with distilled water, the slides were covered with Mowiol and stored in a refrigerator at 4 °C in the dark until evaluation. A BX63 microscope equipped with a DP-80 camera (both from Olympus, Hamburg, Germany) was used for computer-assisted analysis. ImageJ was used to analyze the relative intensities within a specific region of interest (ROI), which defines a part of the hippocampal area CA3. For each animal, the hippocampus of one hemisphere was examined using 4 brain slices. For the analysis of the ROIs, the following parameters were determined: area (µm^2^, the same for each ROI), mean value (average gray value), standard deviation of the gray values, modal value (corresponding to the highest peak in the histogram), the integrated density with IntDen (product of area and mean value) and RawIntDen (sum of the gray values of all pixels). In the control areas, the background intensities were also determined. The R-value, which represents a read-out for the staining intensity depending on the amount of labeled BDNF in the ROI of the section, was calculated next. For that purpose, the individual RawIntDens were set in relation to the control and the relative value (R) was calculated as follows:R = RawIntDen (ROI)/RawIntDen (control) 

Behavioral studies:

Open field (OF): The open-field test is a standardized general measure of motor function (15). A square 45 × 45 cm test arena (Panlab, Cornellà de Llobregat, Spain) with lighting set to 25 lux was used for the OF test. At the beginning of a trial, each animal was placed in the center of the area for 7 min to explore the environment. Tracking began automatically and movements were recorded via a webcam (Logitech C300, Lausanne, Switzerland). Parameters characterizing behavior in the open field (total distance traveled (cm), rearing, resting, and time spent by the animal in the center of the open field) were analyzed from the recorded sessions using SmartJunior 1.0.0.7 (Panlab, Cornellà de Llobregat, Spain). Between trials, the arena was cleaned with 70% ethanol.

Marble burying (MB): The marble burying test is used, for example, as a test for obsessive compulsive disorder [13] and is sensitive to hippocampal dysfunction [14]. Cages were filled with wood chips to a depth of approximately 5 cm and lightly tamped down to create a flat, level surface. A regular pattern of glass marbles was placed on the surface at evenly spaced intervals of about 4 cm each. Each animal was placed in the cage for 30 min. Then, the number of marbles buried to 2/3 of its depth with bedding was counted.

Elevated plus maze (EPM): The EPM consisted of four alternating closed (43.5 × 5 × 14.5 cm) and open (43.5 × 5 cm) arms extending from a small central platform and arranged at 90 degrees to each other. The entire construction was at a height of 43 cm. The arms had a gray plastic floor covered with white plastic coating, the walls of the enclosed arms were 14.5 cm high and made of gray plastic, and the intersections of the four arms formed a 5 × 5 cm open area. At the beginning of the experiment, the mice were placed in an open arm, facing away from the central platform. Rodents get into an attraction–aversion conflict when exposed to a strange stimulus (an open space). Fear and aversion are increased by the elevated, cross-stimulus apparatus. Experimental animals naturally prefer the encircled arms as aversion to the open arms predominates [15].

Dark/light box (DLB): The dark/light box is a tool used to assess anxiety in animals. The fundamental criterion involves observing an animal’s inclination toward dark, enclosed spaces compared to bright, open areas. The dark/light box was divided into a light compartment (35 lux) and a dark compartment. The mice were placed in the light compartment and their behavior was recorded over 7 min using a webcam (Logitech C300, Switzerland). The recorded sessions were analyzed offline as previously described [16].

For statistical analysis, GraphPad Prism version 5 for Windows (GraphPad Software, USA, www.graphpad.com (accessed on 2 February 2024) was used. For the statistical analysis of the data, non-parametric Kruskal–Wallis tests, followed by Dunn’s multiple comparison tests, were used. The level of significance was set to *p* ≤ 0.05. The data presented in the figures (with the exception of Figure 1a) are expressed as mean ± standard deviation (SD). Significant differences are labeled as * *p* ≤ 0.05, ** *p* ≤ 0.01 and *** *p* ≤ 0.001.

## 3. Results

### 3.1. Body and Brain Weight

We first monitored the body weight of the mice. The −/fl,Cre+ mice and the −/+,Cre+ mice gained a higher body weight over time (Figure 1a). By looking at the mean body weight of the mice, it was evident that the +/+,Cre+ mice had the lowest body weight, whereas the −/fl,Cre+ mice displayed the highest body weight (Figure 1b). In detail, the mean body weight (at an age between 6 and 7 months) of the +/+,Cre+ mice was 30.17 g. The mean weight of the −/+,Cre+ mice was 36.71 g. The mean weight of the −/fl,Cre+ mice was 44.63 g. The mean weight of the fl/fl,Cre− mice was 32.87 g, and the mean weight of the fl/fl,Cre+ mice was 36.60 g. In order to obtain a rough idea of the possible alterations in the gross morphology of the brain, we determined the brain weight of adult mice of different genotypes (+/+,Cre+ mice: n = 18; −/+,Cre+ mice: n = 14; −/fl,Cre+ mice: n = 19; fl/fl,Cre− mice: n = 22; and fl/fl,Cre+ mice: n =19). The brain weight did not differ between the groups analyzed, with the exception of the −/fl,Cre+ mice. These mice had significantly smaller brains than the +/+,Cre+, fl/fl,Cre− or fl/fl,Cre+ mice (Figure 1c). The mice with different genotypes also differed in their weekly food consumption, wherein the −/fl,Cre+ mice had the highest food intake and the +/+,Cre+ mice had the lowest food intake (+/+,Cre+ mice: 32.1 ± 2.0 g (n = 19); −/+,Cre+ mice: 35.1 ± 1.7 g (n = 15); −/fl,Cre+ mice: 38.8 ± 3.7 g (n = 18); fl/fl,Cre− mice: 33.3 ± 5.0 g (n = 31); and fl/fl,Cre+ mice: 36.1 ± 7.1 g (n =27)).

### 3.2. BDNF Levels

The BDNF levels were determined indirectly by calculating the R-values for the BDNF immunofluorescence signal in the area CA3 of adult mice (+/+,Cre+ mice: n = 10; −/+,Cre+ mice: n = 10; −/fl,Cre+ mice: n = 4; fl/fl,Cre− mice: n = 3; and fl/fl,Cre+ mice: n = 4). The BDNF-related R-value for the −/+,Cre+ mice, especially the −/fl,Cre+ mice, was the lowest in the area CA3, indicating that these mice expressed the lowest BDNF levels (Figure 1d). Thus, the mice with a higher reduction in brain BDNF showed a higher food intake, high body weight and reduced brain weight. Based on this, we analyzed these mice with regard to their behavior. Since behavioral problems could manifest at different age stages, we analyzed these mice at 3 months, 6 months and 9 months of age.

### 3.3. Behavioral Analysis

Open field: Open-field behavior was analyzed in the five different mouse lines at 3, 6 and 9 months of age (+/+,Cre+ mice [3-month: n = 27; 6-month: n = 21; 9-month: n = 24]; −/+,Cre+ mice [3-month: n = 24; 6-month: n = 15; 9-month: n = 17]; −/fl,Cre+ mice [3-month: n = 15; 6-month: n = 14; 9-month: n = 14]; fl/fl,Cre- mice [3-month: n = 16; 6-month: n = 22; 9-month: n = 22]; and fl/fl,Cre+ mice [3-month: n = 14; 6-month: n = 19; 9-month: n = 19]). When analyzing the distance that the animals traveled, there was no significant difference between the five different mouse lines at three months of age (Figure 2a). At 6 months of age, the running behavior of the −/fl,Cre+ mice significantly differed from the running behavior of the −/+,Cre+ mice. When compared to the other groups, the −/fl,Cre+ mice traveled less than the others, while the −/+,Cre+ mice traveled further than the other groups (Figure 2b). At 9 months of age, the −/fl,Cre+ mice traveled less than the other mouse lines and the −/+,Cre+ mice showed a higher activity (Figure 2c). No major difference was seen between the different mouse lines concerning the times that the mice (irrespective of their age) spent in the center of the arena. No difference in the resting behavior was seen in the young adult mice (3 months of age; Figure 2d). The 6-month-old mice differed in that the −/fl,Cre+ mice displayed significantly longer resting times than the −/+,Cre+ mice (Figure 2e). The analysis of the older (9-month-old) mice revealed that the −/+,Cre+ mice had shorter resting times than the others, while the −/fl,Cre+ mice had the longest resting times (Figure 2f). Concerning jumping behavior, no differences were seen between the different mouse lines at either an age of three, six or nine months. Rearing behavior, however, was different between the mouse lines. The 3-, 6- and 9-month-old −/fl,Cre+ mice differed significantly in their rearing behavior from the other lines (Figure 2g–i).

Marble burying: Burrowing performance can be used to assess brain damage or malfunction that is related to, e.g., neurodegenerative diseases or psychiatric disorders [17]. Marble burying was analyzed in the five different mouse lines at 3, 6 and 9 months of age (+/+,Cre+ mice [3-month: n = 14; 6-month: n = 22; 9-month: n = 22]; −/+,Cre+ mice [3-month: n = 16; 6-month: n = 19; 9-month: n = 19]; −/fl,Cre+ mice [3-month: n = 15; 6-month: n = 14; 9-month: n = 14]; fl/fl,Cre- mice [3-month: n = 12; 6-month: n = 9; 9-month:n = 9]; and fl/fl,Cre+ mice [3-month: n = 16; 6-month: n = 16; 9-month: n = 16]). No differences were detected in the marble burying test (Figure 3a–c).

Elevated plus maze: The elevated plus maze test can be used to analyze anxiety-related behavior and motor activity. Comparable to the previous experiments, we analyzed the five different mouse lines at three, six and nine months of age (+/+,Cre+ mice [3-month: n = 16; 6-month: n = 16; 9-month: n = 16]; −/+,Cre+ mice [3-month: n = 12; 6-month: n = 9; 9-month: n = 9]; −/fl,Cre+ mice [3-month: n = 12; 6-month: n = 12; 9-month: n = 12]; fl/fl,Cre- mice [3-month: n = 15; 6-month: n = 21; 9-month: n = 20]; and fl/fl,Cre+ mice [3-month: n = 14; 6-month: n = 19; 9-month: n = 17]). Concerning their activity, no significant differences were seen between the groups of mice at an age of 3 months (Figure 4a). Interestingly, at 6 months of age as well as at an age of 9 months, the −/fl,Cre+ mice (which represented the group of mice with the lowest level of BDNF in the brain) significantly differed from nearly all groups in that that they traveled less than the others (Figure 4b,c). No differences were seen concerning the number of entries into the open arm or the time that the animals spent in the open arm (for all age groups). In the group of 3-month-old mice as well in the group of 6-month-old mice, sniffing behavior did not differ between the mice of different genotypes.

Dark/light box: Concerning the dark/light box experiments, we analyzed the five different mouse lines at 3, 6 and 9 months of age (+/+,Cre+ mice [3-month: n = 16; 6-month: n = 16; 9-month: n = 16]; −/+,Cre+ mice [3-month: n = 12; 6-month: n = 9; 9-month: n = 9]; −/fl,Cre+ mice [3-month: n = 15; 6-month: n = 14; 9-month: n = 14]; fl/fl,Cre- mice [3-month: n = 15; 6-month: n = 21; 9-month: n = 22]; and fl/fl,Cre+ mice [3-month: n = 14; 6-month: n = 19; 9-month: n = 19]). In the group of 3-month-old mice, the −/fl,Cre+ mice showed the lowest number of entries into the light compartment (Figure 4d). Within the group of 6-month-old mice, the −/fl,Cre+ mice had the lowest number of entries (Figure 4e). Likewise, the 9-month-old −/fl,Cre+ mice displayed a significant lower number of entries when compared to the age-matched mice of the other groups (Figure 4f). In addition, the −/fl, Cre+ mice moved less in the light compartment compared to the +/+,Cre+ (*p* ≤ 0.01), −/+, Cre+ (*p* ≤ 0.001) and fl/fl,Cre- mice (*p* ≤ 0.05). A further parameter that was analyzed was sniffing behavior. The only difference that was noted was that the 3-month-old −/fl,Cre+ mice showed a significantly reduced sniffing behavior when compared to the age-matched +/+,Cre+ mice (*p* ≤ 0.05).

## 4. Discussion

The heterozygous BDNF-knockout mice (−/+,Cre+), the NF-L conditional knockout mice (fl/fl,Cre+) and the newly generated “BDNF double knockout” (−/fl,Cre+) mice were viable and survived into adulthood. In particular, the heterozygous BDNF-knockout mice as well as the “BDNF double knockout” mice displayed a significantly higher food intake when compared to the controls. This was accompanied by a significantly higher increase in body weight over the lifetime, and about 30% of these mice became obese. In this context, it is interesting to mention that increased body weight can be observed in human carriers of the BDNF Val66Met polymorphism [5]. In a recent study, it was shown that youth carriers of the Val66Met BDNF Met-alleles with obesity exhibited significantly increased energy intake when compared to Val-allele carriers, thereby providing support for the possible role of BDNF in appetite, weight and metabolic regulation during adolescence [18]. The reduced availability of functional BDNF might play an important role in obesity. It seems likely that the reduction in BDNF in the brain plays an important role since the conditional BDNF-deficient mice showed an (insignificant) increase in body weight and the “BDNF double knockout” (−/fl,Cre+) showed the highest increase in body weight. This notion is supported by data from McMurphy and colleagues that demonstrate that recombinant adeno-associated virus (AAV)-mediated hypothalamic BDNF gene transfer alleviates obesity and that BDNF gene transfer prevents aging-associated weight gain, improves glucose tolerance, and suppresses inflammatory genes in the hypothalamus and adipose tissues [19].

Being somewhat comparable to heterozygous trkB-deficient mice [20], the total brain weight was reduced in the BDNF mutant mice, especially in those mice with the greatest reduction in available BDNF. As already mentioned, the BDNF Val66Met polymorphism in humans can contribute to reductions in the size of brain areas, e.g., in the hippocampus [6,7] or cortical areas [21]. We noticed that in our mouse models, those mice with the greatest reduction in available BDNF had the lowest brain weight. This may indicate reductions in the size of different brain areas or alterations in the ventricular system. Detailed anatomical analysis of the brain structures or MRI-based analysis of mouse brains might allow us to obtain further insight on this issue. The reductions that may occur are not based on cell loss but on changes in the number or morphology of dendritic spines, and BDNF is known to play a crucial role in the maintenance of hippocampal dendritic spines [4]. Dendritic spines are essential for neuronal signaling and are morphological correlates of neuronal plasticity. Disturbances in dendritic spine morphology can have severe impacts on mental capacities [22]. Such structural alterations might also have an impact on behavior. Thus, we analyzed the different mouse lines by focusing on basic behavior. Forebrain-specific trkB-knockout mice exhibit deficits in hippocampal long-term potentiation (LTP) and hippocampus-dependent learning memory [23], and they display signs of hyperactivity [24]. Being somewhat comparable to this, heterozygous BDNF-knockout mice also display impaired hippocampal LTP [11]. Hippocampal-dependent learning seems to be unaffected in young BDNF heterozygous mice [25], but over time, mice that are heterozygous for BDNF develop age-dependent learning deficits [26]. Furthermore, at least male heterozygous BDNF-knockout mice show signs of increased locomotor activity [27]. We could confirm that the BDNF +/− mice showed higher locomotor activity than the age-matched controls. Interestingly, the “BDNF double knockout” (−/fl,Cre+) mice showed reduced locomotor activity. In most behavioral tests that we performed, the heterozygous BDNF-knockout mice seemed to be more hyperactive, whereas the −/fl,Cre+ mice seemed to be more hypoactive. It could be speculated that this might be linked to the high body weight of the −/fl,Cre+ mice. However, it should be kept in mind that the heterozygous BDNF-knockout mice did not differ significantly in their body weight from the −/fl,Cre+ mice. The difference between these two mouse models is that the heterozygous BDNF-knockout mice had a reduction in BDNF of about 50%, whereas the −/fl,Cre+ mice had an even further reduction in brain BDNF. Thus, the different reduced brain BDNF levels might differentially affect mental driving force and motivation.

The reduced locomotor activity of the −/fl,Cre+ mice is quite remarkable since it has been concluded from other mouse models with reduced BDNF levels that not only heterozygous BDNF-deficient mice can show mild signs of hyperactivity, but that mice with a higher rate of BDNF reduction in the nervous system display hyperactivity and develop anxiety-like behavior [12,28]. In addition, no signs of anxiety-related behavior were observed in the mouse lines examined in this study. No differences in locomotor behavior, when compared to the controls, were seen in the fl/fl, Cre- mice. These data support the data obtained by Andreska and colleagues, who initially generated a conditional mouse model (fl/fl,Cre+) of BDNF deficiency [12].

The mice that had normal BDNF levels (the BDNF+/+,Cre and fl/fl,Cre− mice) did not differ significantly from each other regarding body and brain weights or behavioral parameters. Thus, these two different control strains are both suitable controls for the three different mouse lines that display different reduced levels of BDNF. External factors can modulate brain BDNF. For example, an enriched environment can strongly increase mRNA expression of BDNF in the hippocampus of mice [29]. Furthermore, BDNF is required for the enhancement of hippocampal neurogenesis following environmental enrichment [30]. Thus, it would be interesting to analyze whether there is a threshold level of brain BDNF required to induce the beneficial effects of an enriched environment on the composition and functioning of hippocampal formation. Furthermore, it would be of interest to analyze these different BDNF-knockout models using more complex learning paradigms. Likewise, it might be interesting to analyze whether changes in dendritic spine morphology, dendritic spine densities or dendritic branching patterns might contribute to the changes in gross brain morphology.

## 5. Conclusions

Neurotrophins, like brain-derived neurotrophic factor (BDNF), not only play crucial roles during development but are also involved in postnatal brain processes. Reduced levels of BDNF in mice have an impact on brain size and body weight. In particular, mice with a global heterozygous deficiency for BDNF and an additional strong reduction in brain BDNF displayed a strong gain in body weight over time, and several of these mice were obese. These mice also displayed reduced brain weight. At the basal behavioral level, the BDNF-deficient mice were inconspicuous, with the exception of the mice with a global heterozygous deficiency for BDNF and an additional strong reduction in brain BDNF; these mice travelled less than the other mice. This reduced locomotor activity of the -/fl,Cre+ mice is quite remarkable since other mouse lines with BDNF deficiency often display signs of hyperactivity. Since BDNF is involved in neuronal plasticity, it is likely that these different mouse models of BDNF deficiency display behavioral deficits in more complex learning paradigms, along with changes in dendritic branching patterns, dendritic spines and/or adult hippocampal neurogenesis.

## Figures and Tables

**Figure 1 biology-13-00159-f001:**
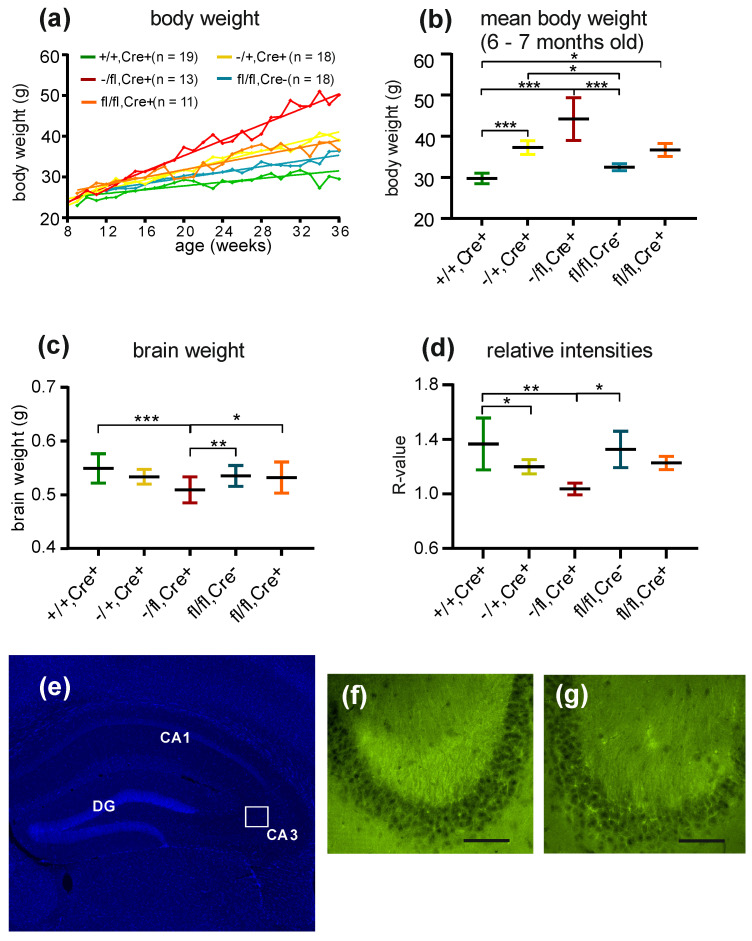
Analysis of body weight, brain wet weight and BDNF levels. (**a**) Body weight was monitored over the period of several weeks. All mouse lines display an increase in body weight over time. This is especially obvious in the case of the −/+,Cre+, fl/fl,Cre+ and −/fl,Cre+ mice. (**b**) The body weight of the different mouse lines was measured at an age between 6 and 7 months and statistically analyzed. This analysis reveals that the −/fl,Cre+ mice have a significantly higher body weight compared to the other mouse lines investigated. (**c**) The brain wet weight was analyzed. The −/fl,Cre+ mice display a lower brain weight when compared to the other mouse lines. (**d**) To analyze the relative BDNF values in the brain, the R-values for BDNF in the area CA3 of the hippocampus were determined. (**e**) Overview of the hippocampus within the areas CA1–CA3 and the dentate gyrus (DG). The rectangle marks the position in the region of interest (ROI) of the intensity measurements. The immunostained area CA3 of (**f**) a +/+,Cre+ mouse and (**g**) a −/+,Cre+ mouse. For better visibility of the stained structures, the images were deconvolved using the “nearest-neighbor” algorithm. The scale bar in f and g: 100 µm. * *p* ≤ 0.05, ** *p* ≤ 0.01 and *** *p* ≤ 0.001.

**Figure 2 biology-13-00159-f002:**
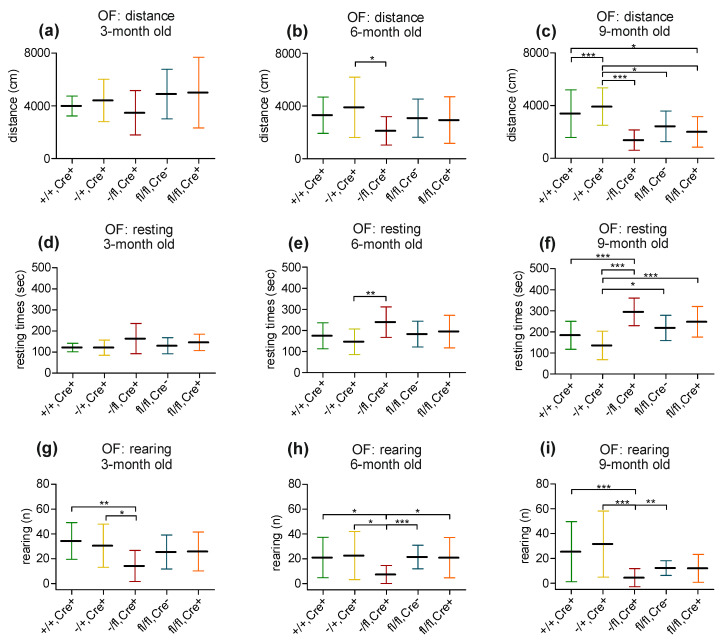
Analysis of data obtained by using an open field (OF). In the OF, the distance that 3-, 6- or 9-month-old mice traveled was determined (**a**–**c**). At an age of 3 months, no obvious differences are seen in this behavior (**a**). In 6-month-old mice, there is a significant differences between the −/+,Cre+ and −fl,Cre+ mice. In general, there is a tendency for the −/+, Cre+ mice to travel further than the other mice and for the −/fl,Cre+ mice to travel less than the other mice (**b**). At an age of 9 months, this behavior is more evident, especially concerning the −/+,Cre+ mice., which travel significantly longer distances than the other mice (**c**). In the OF, resting times were also determined in different groups of 3-, 6- or 9-month-old mice (**d**–**f**). No difference in resting behavior is seen in the young adult mice (3 months of age; (**d**)). The six-month-old mice differ in that the −/fl,Cre+ mice display significantly longer resting times than the −/+,Cre+ mice (**e**). The analysis of the older (9-month-old) mice reveals that the −/+,Cre+ mice have shorter resting times than the others, while the −/fl,Cre+ mice have the longest resting times (**f**). A further parameter analyzed concerns the rearing behavior. At an age of three months, the −/+,Cre+ mice differ significantly in their behavior when compared to the +/+,Cre+ and −/fl,Cre+ mice (**g**). At an age of 6 months, the −/fl,Cre+ mice differ significantly in this type of behavior from all other groups analyzed by displaying reduced rearing behavior (**h**). At an age of 9 months, the −/fl,Cre+ mice still differ significantly from all other mouse lines examined, except for the fl/fl,Cre+ mice that show comparable behavior (**i**). * *p* ≤ 0.05, ** *p* ≤ 0.01 and *** *p* ≤ 0.001.

**Figure 3 biology-13-00159-f003:**
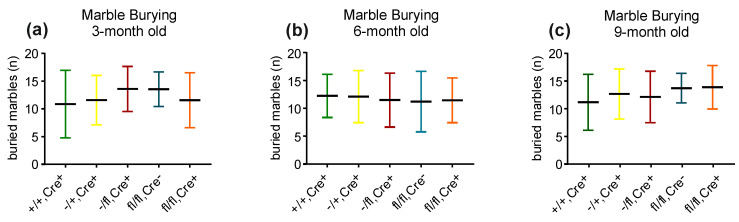
The analysis of marble burying behavior did not show differences in the behavior of the different groups of mice at either an age of three (**a**), six (**b**) or nine months (**c**).

**Figure 4 biology-13-00159-f004:**
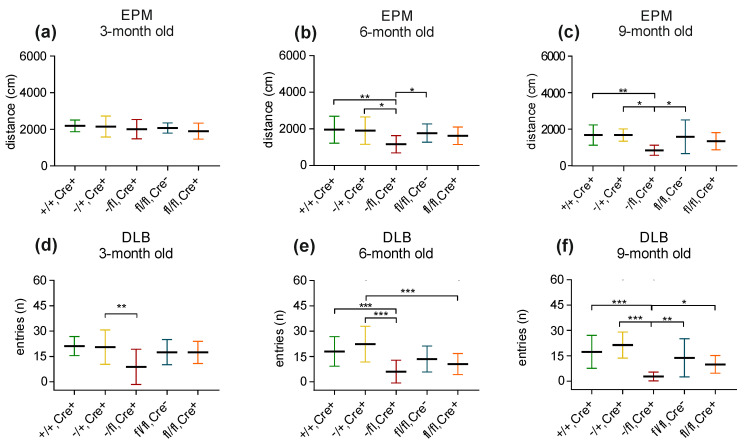
Elevated plus maze (EPM) and dark/light box (DLB). In the EPM, there is no difference in the distance that the 3-month-old mice travelled (**a**), whereas in the group of 6-month-old mice (**b**) as well as in the group of 9-month-old mice (**c**), the −/fl,Cre+ mice show reduced traveling behavior. In the DLB, the 3-month-old −/+,Cre+ mice show more entries than the age-matched −/fl,Cre+ mice (**d**). In the group of 6-month-old mice, the −/fl,Cre+ mice display the lowest number of entries, whereas the −/+,Cre+ mice show the highest number of entries into the other compartment (**e**). When compared to all groups of 9-month-old mice, the −/fl,Cre+ mice display the lowest number of entries (**f**). * *p* ≤ 0.05, ** *p* ≤ 0.01 and *** *p* ≤ 0.001.

## Data Availability

The data presented in this study are available from the corresponding author upon request, subject to restrictions.

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
