# Peer review of "Reduced Levels of Brain-Derived Neurotrophic Factor Affect Body Weight, Brain Weight and Behavior"

_biology, 2024, doi:10.3390/biology13030159_

Round 1
Reviewer 1 Report
Comments and Suggestions for Authors
My comments are reported in the attached PDF file.

Author Response
We wish to thank the reviewers for their time and their effort invested for reviewing our manuscript.
We have tried to revise our manuscript accordingly to the suggestions made by the editorial office and the reviewers.
Editorial office: General issues:
We have reduced the number of self-citations by deleting Reference 22 and 25.
We have added the date of the ethical approval
We have added a conclusion
Rev: #1
1) Thank you for the kind comments.
2) Indeed, Figure 9 is not present in the manuscript, since the figures have been re-arranged., Fig. 9 has been deleted from the manuscript
3) Yes, indeed, there is also a mistake in the numbering of the figures at line 256 it should be Fig. 4 instead of Fig. 3; we have corrected this issue; we also corrected Figure 3 to Figure 4 on page 8. We also corrected”3, 3 and 9 months of age” to “ 3, 6 and 9 months of age”.
4) Thank you for noting this issue concerning the citation. We have corrected these sentences.
5) We have added a conclusion to the end of the manuscript.
6) We have checked the manuscript and corrected some further typos.
Reviewer 2 Report
Comments and Suggestions for Authors
This study conducted by Voigt et.al measured the body weight, brain weight and behavior of mice with different BDNF level. The authors generated a new mouse strain where neurofilament L positive neurons express no BDNF and other cells have heterozygous BDNF depletion. It is interesting that the –/fl,Cre+ mice showed many behavioral phenotypes that are not present in heterozygous deficient mice.
Although it is nice to have side by side comparison of mice with different BDNF level, this study did not provide novel findings on the impact of BDNF deficiency on body weight or brain weight. It is known that deletion of BDNF in the whole brain or the hypothalamus causes hyperphagia and obesity in mice. Although they characterized the behavior changes in these mice, they did not provide any mechanistic connection of the reduced BDNF to these phenotypes.
Figure 1. the authors should provide representative images of the mouse brain and body to show the size change, and immunostaining images for panel d.
Page 4, line 181, the authors need to present data to support ‘higher reduction of brain BDNF showed a higher food intake’. Increased body weight could be from increased food intake and/or reduced metabolism.
Page 5. The authors should provide the rationale of choosing the open field and dark light box behavior tests.
Page 1, line 10. This sentence is hard to read: ‘Mice are available that have less BDNF than normal.’
Figure 4 is mislabeled as Figure 3.
Comments on the Quality of English Language
The simple summary part needs some polishing.
Author Response
Editorial office: General issues:
We have reduced the number of self-citations by deleting Reference 22 and 25.
We have added the date of the ethical approval
We have added a conclusion
Rev: #2
Comment:
Figure 1: Unfortunately, we cannot present representative images of the mouse body size, since the mean values are based on measurements of a lot of individual animals over a large time scale. In addition, the mice with deletions in the BDNF system are highly variable in their bodyweight. We have currently not enough data for a in-depth analysis and appropriate statistical evaluation. At least, we noted that – in general –the male mice have a higher body weight than females. We further noted that the group of the BDNF-deficient mice is inhomogeneous, with “normal” weighted mice and mice that show signs of being obese. Therefore, we have defined a cut off point (COP) for obesity: “Cut Off Point =mean weight (of the control mice) +standarddeviation x 2”, a BDNF-deficient mouse was considered of being obese if the mouse has in three following measurements a higher body weight as the COP. According to this, several of the BDNF deficient mice are getting obese over time:
Genotyp BDNF-/+, Cre+ BDNF-/fl, Cre+ BDNFfl/fl, Cre+
♂ ♀ ♂ ♀ ♂ ♀
(n) 11 4 10 9 7 10
Obese
(in %) 8
(73 %) 0
(0 %) 7
(70 %) 5
(56 %) 2
(29 %) 2
(20 %)
However, currently we have not enough single data for doing adequate statistics and cannot present these data. At least these data hint that the BDNF deficient mice have a tendency for getting obese over time and, in addition, the sex of the animals seems also to have an impact.
Concerning representative images of the different brains We have tried to make photographs that clearly show these differences. However, this difference cannot clearly be seen in a photograph of five different brains (each representative for a genotype). This might be due to the small difference: The lowest mean brain weight of one group was 0.509 and the highest mean brain weight was 0.549 in another group, which is a difference of about 7 %. However, an increase in brain size must not be equivalent to an increase in brain weight as e.g. in case of a hydrocephalus.
Concerning the immunostainings We have added an image giving an overview of the hippocampus and showing the position of the ROI (DAPI was used to visualize cell nuclei (in blue)). In addition, we present two examples of the immunostaining in area CA3 of a BDNF +/+ and a BDNF -/+ mouse. For this presentation, we deconvolved the images using nearest-neighbor algorithm, since it allows a better visibility of the stained structures. The nearest neighbor algorithm was not used for the intensity-measurements.
Page4, line 181: Indeed, higher body weight might be due to reduced metabolism and/or increased food intake. With our equipment, we are not able to monitor the metabolism of the individual mice for several weeks. However, we have data concerning their food intake. We added thisto the manuscript: in the section materials and methods we added: “To determine the weekly feed quantity, a beaker was placed on a standard kitchen scale and zeroed. The food was then poured into the beaker and a weekly average was calculated from the difference between the amount of food added in the previous week and the amount in the trough on the day of measurement.” And we added to the section results: “The different genotypes also differ in their weekly food consumption, whereby the -/fl,Cre+ had the highest food intake and the +/+,Cre+ mice had the lowest food intake (+/+,Cre+ mice: 32.1 ± 2.0 g (n = 19); -/+,Cre+ mice: 35.1 ± 1.7 g (n = 15); -/fl,Cre+ mice: 38.8 ± 3.7 g (n = 18); fl/fl,Cre- mice: 33.3 ± 5.0g (n = 31); fl/fl,Cre+ mice: 36.1 ± 7.1 g (n =27)).”
Page 5: we have added why we in the section “Materials and methods”, for what the the open field test and the dark light box test are used.
Page 1, line 10: “Mice are available that have less BDNF than normal” -> We have re-written this paragraph.
Figure 4 is mislabeled as Figure 3. -> We have corrected this issue.
Simple summary need some polishing: -> We have rewritten some parts of the “Simple summary”
Round 2
Reviewer 2 Report
Comments and Suggestions for Authors
I am generally satisfied with the revisions. Although some of the concerns are not directly addressed experimentally, their textual responses addressed the major concerns.